# Diet and Skin Aging—From the Perspective of Food Nutrition

**DOI:** 10.3390/nu12030870

**Published:** 2020-03-24

**Authors:** Changwei Cao, Zhichao Xiao, Yinglong Wu, Changrong Ge

**Affiliations:** 1Livestock Product Processing Engineering and Technology Research Center of Yunnan Province, Yunnan Agricultural University, Kunming 650201, China; ccwylf1111@163.com (C.C.); jtf413171@126.com (Z.X.); 2College of Food Science, Sichuan Agricultural University, Ya’ an, Sichuan 625014, China; wuyinglong99@163.com; 3College of Food Science and technology, Yunnan Agricultural University, Kunming, Yunnan 650201, China

**Keywords:** diet, skin aging, nutritional level, eating habits, foodborne antioxidants, anti-aging

## Abstract

We regularly face primary challenges in deciding what to eat to maintain young and healthy skin, defining a healthy diet and the role of diet in aging. The topic that currently attracts maximum attention is ways to maintain healthy skin and delay skin aging. Skin is the primary barrier that protects the body from external aggressions. Skin aging is a complex biological process, categorized as chronological aging and photo-aging, and is affected by internal factors and external factors. With the rapid breakthrough of medicine in prolonging human life and the rapid deterioration of environmental conditions, it has become urgent to find safe and effective methods to treat skin aging. For diet, as the main way for the body to obtain energy and nutrients, people have gradually realized its importance to the skin. Therefore, in this review, we discuss the skin structure, aging manifestations, and possible mechanisms, summarize the research progress, challenges, possible directions of diet management, and effects of foodborne antioxidants on skin aging from the perspective of food and nutrition.

## 1. Introduction

Skin is the organ with the largest contact area between the human body and the external environment and is a barrier that separates the human body from the environment. It not only protects the body from external environmental damage and avoids water loss from the body, but also has a certain cosmetic effect [1]. The aging of organs occurs throughout our life. As the largest organ of the human body, the skin shows obvious signs of aging due to age, ultraviolet radiations (UVR) exposure, and chemical pollution. With the development of science and technology and improvement in human living standards, people pay more attention to skin aging and try to have a better understanding of it. Many people, especially women, spend a significant portion of their daily expenses on cosmetics and medicines for the treatment and prevention of skin aging. This huge demand continues to drive research into the prevention and treatment of skin aging [2].

Animal skin is composed of three layers, including epidermis, dermis, and subcutaneous tissue [3] (Figure 1). During development, skin epidermal cells rapidly differentiate into four layers of stratum corneum, granular layer, spinous layer, and the basal layer. Stem cells (SC) and transient amplification cells (TA) located at the base layer promote the regeneration of human skin epidermis. Epidermal regeneration and SC behavior are regulated by external signaling pathways such as the Wnt signaling pathway [4,5]. Dermis refers to the part above the subcutaneous fat below the epidermis, which is the connective tissue composed of fibroblasts, responsible for the synthesis and secretion of collagen and other matrix proteins (such as fibronectin, elastin, and glycans) to the extracellular environment, giving the skin elasticity, strength, and ability to resist external interference [6,7]. Fibroblasts are also involved in skin aging [8], carcinogenesis [9], wound healing [10], fibrosis [11], and other pathological processes. Subcutaneous layer refers to the fat layer immediately below the dermis layer, which surrounds the hair follicles and plays a major role in connecting the skin with muscles and bones, storing energy, secreting hormones, and keeping warm. Subcutaneous adipose tissue is also involved in regulating the speed of hair regeneration, balancing the internal environment of the skin, and promoting skin repair after damage and infection [12,13].

Aging refers to the body’s ability to adapt to the environment’s physiology and psychology, which progressively decreases and gradually leads to death. It is mainly characterized by the accumulation of macromolecular damage, impaired tissue renewal, gradual loss of physiological function integrity, and increased risk of death [14]. Aging is caused by a combination of internal factors (such as hormone levels, genotypes, endocrine metabolism, etc.), and external factors (such as ultraviolet radiation, nutritional levels, chemical pollution, etc.). Skin aging can be divided into chronological aging and photo-aging (or internal aging and external aging). As the name suggests, the chronological aging of the skin occurs throughout the body, and photo-aging occurs on the body’s light-exposed sites [15,16]. Chronological aging caused by internal factors occurs naturally and is not easy to change, but it is possible to delay photo-aging by altering external factors [17]. Reasonable diet and balanced nutrition are important measures to delay aging and prolong life. Therefore, in this review, we briefly introduce the composition of the skin, internal and external changes that occur during skin aging, and the molecular mechanism, focus on the perspective of food nutrition, and review the recent progress in research on diet management, nutrition regulation, and foodborne antioxidants in inducing and delaying skin aging. We hope that our views will enrich the reader’s understanding of diet to improve skin aging and also provide a basis for subsequent research.

## 2. Changes and Molecular Mechanisms in Skin Aging

### 2.1. Apparent Changes in Skin Aging

Aging is characterized by the accumulation of macromolecular damage within cells, impaired ability of stem cells to promote tissue regeneration, and restore the loss of physiological integrity [7]. Chronological aging and photo-aging are two processes of skin aging that although related, have different clinical manifestations and pathogenesis (Figure 2). Chronological aging usually appears after a certain age and is affected by factors such as ethnicity, individual, and skin site. It is mainly characterized by dry skin, dullness, lack of elasticity, and fine wrinkles [18]. Histological features include epidermal atrophy, reduction in the number of dermal fibroblasts and collagen fibers, slackening, thinness, and even function disorganized. The primary causes are: first, the SC dysfunction in keratinocytes, decreased regenerative ability of stem cells in the basal layer of the epidermis leading to a decline in skin renewal and repair ability, ultimately causing aging [19], and second, due to the accumulation of damage and aging skin dysfunction, fibroblasts lose the ability to reshape the extracellular matrix or have a reduced ability to synthesize and secrete collagen or viscous proteins. Third, aging fibroblasts alter intracellular homeostasis through certain paracrine mechanisms [20,21].

Photo-aging is caused by long-term exposure to ultraviolet radiation, mainly manifested as skin wrinkles, relaxation, roughness, yellowish or grayish-yellow, capillary expansion, and pigmented spot formation, etc. [16,19]. The photo-aging site is affected by the wavelength of ultraviolet light, which can be divided into A, B, C categories depending on the length of the wavelength. UV-A (320–400 nm) has low energy but strong penetrating power, which mainly affects the dermis of the skin. It accelerates the hydrolysis of skin collagen by promoting the production of matrix metalloproteinases (MMPs), leading to tissue destruction and progressive degeneration of dermal extracellular matrix [22,23,24]. UV-A also inhibits the synthesis of hyaluronic acid (HA) by down-regulating the synthesis of hyaluronic acid synthase, thereby changing the composition of skin proteoglycans [20]. UV-B acts on keratinocytes in the skin epithelial layer, potentially induces DNA damage and mutation in keratinocytes, stimulates the release of soluble cytokines from keratinocytes, and causes skin cells to show symptoms such as aging, inflammation, apoptosis, and carcinogenesis [25,26]. However, keratinocytes have a strong antioxidant capacity in response to UV-B exposure, are more resistant than fibroblasts to the lethal effects of oxidants, and are more sensitive to reactive oxygen species (ROS)-induced apoptosis [27]. Further, aging skin fibroblasts cause pigmentation and dark spots by promoting the transcription of the melanin gene [28].

### 2.2. Molecular Mechanism of Skin Aging

Researchers have used many models in recent years to explain the molecular mechanism of skin aging and the mechanism of its alleviation. These models include cell aging, oxidative stress, high-frequency chromosomal abnormalities, single-gene mutations, and chronic inflammation [29]. We summarize here the progress in recent years on the research on the molecular mechanism of skin aging, as fellows.

a.Oxidative stress. Oxidative stress plays an important role in skin aging and skin damage processes, and its main feature is increased intracellular ROS. The skin’s oxidative metabolism and UV exposure lead to the production of ROS. The accumulation of ROS causes DNA damage, induces skin inflammatory response, reduces antioxidant enzymes, activates nuclear factor kappa B (NF-kB) and activator protein1 (AP–1) to inhibit collagen production, and increases matrix metalloproteinases to decompose collagen and binding proteins in the dermis, which eventually leads to skin aging [30,31,32,33].b.DNA damage and gene mutation. Earlier studies reviewed the mechanisms of UV-induced DNA damage and classified them into direct damage and indirect damage. Direct damage occurs when DNA absorbs the UV-B photon, leading to rearrangement in the nucleotide sequence, resulting in DNA strand deletion or mutation. During indirect damage, DNA molecules absorb UV-A and promote electron and energy transfer to oxygen molecules to form free radicals’ singlet oxygen ions, causing DNA damage [34,35]. DNA damage can be repaired by photolytic enzymes, while UV-induced skin DNA damage can be prevented by applying sunscreen [36,37].c.Shortening of the telomere. Telomeres are a small piece of DNA-protein complex at the end tips of eukaryotic linear chromosomes, which are important components in maintaining the chromosomal integrity and controlling the cell cycle. A telomere is shortened with cell division and is associated closely with cell division and senescence [38,39]. Telomerase is an enzyme responsible for telomere elongation, and its synthesis is essential for telomere maintenance and long-term survival of the organism. Epithelial stem cells with short telomeres have a poor proliferative capacity, which can be corrected by introducing telomerase. The reactive oxygen generated by UV radiation induces telomere mutation, cell death, or senescence. Nevertheless, some studies opine that the relationship between telomeres and aging may be speculation and, therefore, this relationship needs to be demonstrated [38,39,40,41,42].d.The role of microRNA. miRNAs are a type of conserved non-coding RNA. Chronic UV-B alters the expression of mir–34 family proteins in the skin. MiR–34 in human dermal fibroblast (HDFs) cells regulates cell function and expression of MMP–1, α1 type1 collagen (COL1A1), and elastin. *miRNA 378b* inhibits mRNA expression of COL1A1 by interfering with Sirtuin 6 (SIRT6) in HDFs, miRNA 217 regulates the senescence of human skin fibroblast by directly targeting DNA methyltransferase 1, and miR–23a–3p controls cellular senescence by targeting enzymes to control hyaluronic acid synthesis. These studies thus show that microRNAs regulate the skin aging process [43,44,45,46,47].e.Accumulation of advanced glycation end products (AGEs). AGEs are the products of excess sugar and protein binding, usually derived from body synthesis and food intake. The nonenzymatic glycosylation aging theory has been widely recognized by many scholars. As the final product of nonenzymatic glycosylation reaction, AGEs accumulate in photo-aging skin, affect protein function in the dermis, and promote skin aging [48,49,50].f.Aging due to inflammation. Continuous UV radiation exposure induces oxidative stress in epidermal cells, causing cell damage, fat oxidation, and finally leads to cell inflammation. When the degree of inflammation exceeds the ability of macrophages to clear up, macrophages also begin secreting pro-inflammatory factors and ROS to accelerate dermal inflammation and injury [51,52]. With the continuous advancement in dermatology in the past two decades, methods such as stem cell transplantation, hormone therapy, telomerase modification, and use of antioxidants and retinoic acid have been promoted to address skin aging. However, some of these treatment methods have certain disadvantages and serious side effects. For example, hormone therapy increases the risk of breast cancer, retinoic acid may cause osteoporosis, and telomerase modification increases the risk of skin cancer. Therefore, improving skin condition through diet management is being increasingly accepted by people.

## 3. Diet Management and Skin Aging

Food is the foundation of our lives, and diet is the main way for the body to obtain the required substances for growth and maintenance. Human beings group themselves into different ethnicity, religions, nationalities, and catering cultures. More than 2000 years ago, the Chinese medical book "Yellow Emperors Internal Classic. Su Wen" contained a balanced diet principle of "five grains for nutrition, five fruits for help, five animals for benefit, and five vegetables for filling", and the folk also includes "what to eat and what to add". Modern science has proven that an imbalance in nutrition and poor eating habits are important causes of skin aging. The effect of nutrients and dietary habits on skin aging is mentioned in Table 1.

### 3.1. Nutrition Level

Nutrition is closely associated with skin health and is required for all biological processes of skin from youth to aging or disease. Nutrition levels and eating habits can repair damaged skin and can also cause damage to the skin. In recent years, a number of people have closely linked health-nutrition-eating habits and skin health, besides, clinical research and epidemiology have successfully combined nutrition with tissues and organ health and have confirmed that nutritional levels and eating habits have a certain degree of impact on skin health and aging.

Water is a vital constituent of the body and facilitates maintenance of balance and tissue function in the body. Water in the body and cells mainly serves the role of nutrient, solvent, transportation carrier, maintains body volume, and regulates body temperature [53,54,55]. Lack of water in the body can cause tissue dehydration and functional disorders (such as aging and inflammation). Skin is no exception, and the appearance of the skin on lips and limb is a direct reflection of the body’s moisture status. So how much water every day is good for the skin? Studies show that it is better to have more water and drinking more than 2 L of water per day significantly affects skin physiology and promotes superficial and deep hydration of the skin. However, the effects of water on the skin may be different from that of the water intake, and these effects are obvious in people who drink less water [56,57].

Trace elements include iron, iodine, zinc, and copper, etc., and refer to elements whose content in the human body is less than 0.01–0.005% of the body mass. Despite being less abundant in the body, trace elements have strong physiological and biochemical effects [58]. Trace elements are closely related to skin immunity and inflammation, and the homeostasis of copper and zinc ions in psoriasis patients may be a potential target for treating psoriasis [59]. Zinc content in the skin ranks third among all tissues and is an essential element for the proliferation and differentiation of skin epidermal keratinocytes [60,61]. Bauer et al. [62] demonstrated that enhanced dietary zinc-rich amino acid complexes may affect the proliferation of goat horn and interphalangeal skin keratinocytes. However, the role of trace zinc or amino acid has not been further clarified. In the skin, copper is involved in the extracellular matrix formation, synthesis and stabilization of skin proteins, and angiogenesis. Clinical studies have shown that copper aids in improving skin elasticity, reducing facial fine lines and wrinkles, and promoting wound healing [63]. Iron is a catalyst for bio-oxidation. Studies have shown that ultraviolet radiation and iron content in women’s post-menopausal skin cells increase rapidly, reduce the skin’s antioxidant capacity, and lead to aging [64,65]. The lack of selenium in the diet weakens the UV-B-induced antioxidative ability of mice skin, making the skin more sensitive to oxidative stress due to ultraviolet radiation [66]. Se-enriched proteins are also essential in keratinocyte development and function [67]. There are some studies on the lack of other trace elements and their effects on the skin, as well as in vitro experiments of trace elements, which are not described here.

Vitamin deficiency affects skin health. The lack of vitamins in the body can cause skin disorders. For example, lack of vitamin C causes the symptoms of scurvy such as fragile skin and impaired wound healing. Vitamins, as skin antioxidant defense ingredients, are mostly taken from food, so the content of vitamins in the diet is closely related to skin antioxidant capacity and physiological functions [68,69,70].

Proteins form an important part of body tissues and organs. Their primary physiological functions are to construct and repair tissues, mediate physiological functions, and supply energy. All tissue cells in the body are constantly renewed, and only adequate protein intake can maintain normal tissue renewal and repair. Skin is no exception, and the skin renewal cycle is generally considered to be 28 days. Protein deficiency or excessive intake can cause metabolic disorders and affect physical health [71]. Excessive intake of plant protein increases kidney load, and excess animal protein intake increases the risk of osteoporosis [72,73]. Conversely, protein deficiency causes a series of diseases such as reduced resistance, slow growth, weight loss, apathy, irritability, anemia, thinness, and edema. Countries around the world recommend standard protein intakes based on age, sex, work, and physiological period. Studies have shown that ingesting sufficient protein can help in healing pressure ulcers in rats, while both excess and inadequate protein intake are detrimental to ulcer healing. Dietary protein supplementation can also enhance cellular protein synthesis and metabolism [74]. The effects of dietary protein and its hydrolyzed peptides improving skin aging have been described in detail in Section 4.1 of this paper. There may be some other nutrients that have not been discussed in this article but affect the skin aging and needs to be continuously supplemented.

### 3.2. Eating Habits

Dietary habits refer to the preference for food or drink, are an important part of the dietary culture and influenced by regional, historical, cultural, product, and other factors. Although the incidence of vitamin, trace elements, and protein deficiencies in developed western countries are very low, imbalanced or incomplete diet can also lead to diseases and aging, thereby affecting skin health. Data from epidemiological and experimental studies suggest an important role of diet and dietary patterns in the pathogenesis of many age-related diseases [75].

Tobacco use is one of the major public health hazards in the world. Millions of people worldwide die each year due to smoking, so tobacco is also called a "poisonous weed". Smoking can change skin cuticle thickness and accelerates skin pigmentation. The thickness of the stratum corneum correlates positively with pigmentation and negatively with years of smoking, and this skin pigmentation is more obvious in the upper lip than in the gums [76,77]. Some clinical observations and investigations have also shown a certain correlation between smoking, external aging, and facial skin aging [78,79]. Further, after cosmetic surgery, smoking can cause complications such as postoperative infections, delayed wound healing, and skin necrosis [80]. While we cannot conclude completely on the harmful effect of alcohol on the body, alcohol and acetone produced by alcohol metabolism can promote the proliferation of skin keratinocytes, thereby enhancing skin permeability and damaging its barrier function. Alcohol also affects the metabolism of triglycerides and cholesterol and affects the lipid composition of the skin [81,82]. Studies by Goodman et al. [83] revealed that aging causes changes in facial skin and volume and is closely related to smoking and heavy drinking, and the degree of facial aging increases with the amount and time of exposure to tobacco and alcohol. On the contrary, quitting smoking and alcohol can delay the aging of facial skin. Dysfunction of alcohol metabolism in the aldehyde dehydrogenase 2 *(ALDH2)* gene knockout mouse or human allele also confirmed that alcohol can cause increased skin pigmentation, although the downstream mechanism of action is unclear [84]. While there are some studies on the relationship between alcohol intake and skin diseases, the relationship between alcohol intake and disease needs to be accurately determined based on real situations [85].

A high-fat diet is closely related to various diseases such as obesity, diabetes, fatty liver, and skin aging. Raman spectroscopy studies have shown that dietary fat intake is closely related to the body’s adipose tissue and the lipid composition of the skin [86]. High-fat diets delay healing of the skin by promoting skin oxidative stress and inflammatory responses, reducing protein synthesis, and may also cause morphological changes in skin and damage to matrix remodeling [87,88]. Psoriasis is a systemic chronic skin inflammatory disease that affects 2–5% of the population in western countries. The free fatty acids content in serum is an important parameter to reflect the severity of obesity-related diseases. A high-fat diet led to an increase in free fatty acid content in mice, which is an important factor that aggravates skin inflammatory psoriasis. Moreover, a high-fat diet can promote skin inflammation and cancer by enhancing the expression of inflammatory factors and tumor necrosis factor in the skin by UV-B [89,90,91]. Studies by Zhang and others [92] showed that after high-fat diet, the epidermal fatty acid-binding protein (E-FABP) of mice was significantly upregulated in the skin, which promoted the formation of lipid droplets and the activation of NLRP3 inflammator, and greatly increased the incidence of skin lesions in mice. In general, the effect of a high-fat diet is mainly to cause aging of the skin by causing skin oxidative stress to produce inflammatory damage.

Some studies have also shown a close association between sugar and some food processing methods (such as grilling, frying, baking, etc.) with skin aging, and their mechanisms are related to skin advanced glycation end products. A high-sugar diet, ultraviolet irradiation, and eating barbecued fried foods, lead to the accumulation of AGEs and acceleration of skin aging. However, strict control of blood sugar for four months can reduce the production of glycosylated collagen by 25%, and low-sugar food prepared by boiling can also reduce the production of AGEs [93,94,95]. When the mice were fed with carbohydrate-controlled diets for 50 weeks, the epidermis and dermis were significantly thinned, autophagy was inhibited, and inflammation was exacerbated. Mechanistically, long-term intake of carbohydrates in mice promotes skin aging by activating the mammalian target of rapamycin (mTOR) [96]. Further, high-salt, spicy, and extremely vegetarian diets are also considered to be detrimental to skin health. Therefore, scientific, reasonable, healthy, and diverse eating habits and eating some antioxidant-rich foods are essential to maintaining skin health.

**Table 1 nutrients-12-00870-t001:** Summary of key effects of nutrients and diet in skin aging.

Nutrients/Diet	Relationship with the Skin	References
Water	Maintain skin internal balance and tissue function (e.g., aging and inflammation)	[56,57]
Proteins	Constitution and repair of skin tissues (involved in protein synthesis and metabolism), mediation of skin physiological functions and supply of energy.	[74]
**Trace Elements**		
Copper	Involved in extracellular matrix, synthesis and stabilization of skin proteins, and angiogenesis.	[63]
Zinc	Participates in the proliferation and differentiation of epidermal keratinocytes.	[60,61]
Iron	Closely related to the activity of antioxidant enzymes in skin cells.	[64,65]
Selenium	1. Essential for the development and function of skin keratinocytes.2. Related to skin antioxidant enzyme activity.	[67]
**Vitamins**		
VA	Commonly used anti-aging ingredients prevent skin aging by regulating the expression of genes and matrix metalloproteinases.	[97,98]
VB	Associated with skin inflammation and pigmentation.	[99]
VC	Involved in skin collagen synthesis and elimination of intracellular reactive oxygen species.	[100]
VD	Reduces skin DNA damage, inflammation, and photocarcinogenesis.	[101]
VE	Prevent skin aging by inhibiting lipid peroxidation.	[102,103]
**Diet**		
Fat	High fat is associated with skin inflammation, essential fatty acids are involved in skin lipid synthesis and metabolism.	[92,104]
Tobacco	Change skin cuticle thickness, accelerate skin pigmentation and skin necrosis.	[76,77,78,79]
Alcohol	Promote the proliferation of keratinocytes, change the skin permeability, destroy the barrier function of the skin, affect the skin lipid composition.	[81,82]
Sugar and baked goods	Associated with skin thickness, AGEs, autophagy, and inflammation.	[93,94,95,96]

## 4. Foodborne Antioxidants and Skin Aging

According to free radical theory, lipid peroxidation, DNA damage, and inflammation are the primary causes of skin aging, disease, and dysfunction. This led to a medical revolution that emphasized antioxidants and free radical scavengers for the prevention and treatment of skin aging [105,106]. Oxygen-free radicals are ubiquitous in the process of cell metabolism and can interact with DNA, proteins, and polyunsaturated fatty acids in the body, causing breaks in the DNA chain and oxidative damage, protein–protein cross-linking, protein–DNA cross-linking, and lipid metabolism oxidation, etc. ROS can cause various cardiovascular diseases, cancers, and aging. In vivo oxidation eventually leads to aging of the organism, so exogenous antioxidant supplements, with food being an important source, have become a topic of research [107]. Therefore, this section lists the key studies on the effect of natural antioxidants such as collagen peptides, polyphenols, vitamins, polysaccharides, and fatty acids extracted from foods on alleviating skin aging (Table 2). Some antioxidants are also extracted from food, and their effect of alleviating skin aging is studied through local skin penetration in vitro. However, they then do not belong to diet, hence, they are not summarized here.

### 4.1. Collagen Peptide

Collagen is a long cylindrical polymeric protein, the main component of animal extracellular matrix (EMC), and the most abundant and widely distributed functional protein in mammals, accounting for 25% to 30% of the total protein; some organisms constitute even up to 80% or more collagen, which have unique physiological functions and are widely used in food, medicine, tissue engineering, cosmetics, and other fields [108]. Traditionally, collagen is thought to improve skin health, but later research found that collagen peptides with smaller molecular weight are easier to absorb and have more significant effects. While collagen is mainly absorbed and utilized in the form of peptides in mice, with a utilization rate of about 50%, collagen peptides can be almost completely absorbed and utilized by the body [109]. Collagen peptide is a series of small molecular peptides obtained from the proteolytic hydrolysis of collagen. Because of their small molecular weight, easy absorption, anti-inflammatory, and antioxidant characteristics, collagen peptides have been studied as a new favorite in recent years as exogenous antioxidants to relieve skin aging. Collagen peptides mainly come from animal skin, bones, tendons, muscles, and other tissues. The research on animal-derived collagen peptides and other protein peptides in alleviating skin aging in recent years is summarized in Table 2. Nevertheless, due to associated risk of diseases such as mad cow disease, foot-and-mouth disease, etc., and religious controversy of animal-derived collagen peptides, in recent years, people have also evaluated the effects of non-collagen peptides such as walnut protein peptides and whey protein peptides on alleviating skin aging in mice [110]. There are also some cases about the combined application of protein peptides and other nutrients, although, they have not been summarized because the functions of various nutrients cannot be distinguished.

Mechanistically, collagen peptides and other protein peptides may relieve skin aging through either of the three pathways. First, the protein or peptide enters the blood circulation after digestion and absorption, and then participates in the skin fibroblasts as a precursor of collagen synthesis, thereby protecting the aging skin. Second, collagen peptides that enter skin cells incur anti-aging effects by removing ROS from cells, protecting the cell’s endogenous antioxidant defense system, and reducing oxidative damage and inflammatory responses in cells. In the third pathway, protein peptides entering skin cells promote collagen and hyaluronic acid synthesis and inhibit the production of inflammation by regulating cytokines and activating TGF–β/Smad or other signaling pathways, while, these peptides concurrently also prevent skin collagen degradation by inhibiting the expression of proteases such as nuclear transcription factor activating protein–1 (AP–1), MMP–1 and MMP–3.

### 4.2. Polyphenols

Polyphenols are secondary metabolites of plants and exist widely in vegetables, fruits, tea, and other plants. Due to their obvious antioxidant properties, polyphenols have become one of the most important compounds to be used in cosmetics and nutritional cosmetology to combat skin aging. In recent years, tea polyphenols, curcumin, flavonoids, silymarin, and grape resveratrol have been the most studied polyphenols with anti-aging properties. Polyphenols reduce oxidative damage and inflammation in the skin through their antioxidant and anti-inflammatory effects, mainly by inhibiting collagen degradation, increased collagen synthesis, and inhibiting inflammation, which involves the regulation of matrix metalloproteinases, cytokines, and signaling pathways (e.g., Nrf2, NF-κB, MAPK, etc.) [111,112]. This article lists some experimental cases of food-derived polyphenol extracts to alleviate skin aging (Table 2), but most of the anti-aging effect of polyphenols has been verified in vitro through topical administration on target skin cells. Nonetheless, the clinical application of polyphenols in dermatology is still in its infancy, and there are relatively few reports on the cytotoxicity of polyphenols [113,114].

### 4.3. Polysaccharides

Polysaccharides are polymer carbohydrates formed by the dehydration and condensation of multiple monosaccharides. Due to their pharmacological effects, such as improving immune function, anti-tumor, anti-virus, anti-glucose, anti-oxidative, lowering blood lipids, and low cytotoxicity, polysaccharides are an ideal functional food and drug active ingredient [115]. Polysaccharides have been the research focus from the past five years, and there are several reports on various types of polysaccharides. In general, the research mainly focuses on the isolation and extraction of polysaccharides, structural identification, modification, and determination of physiological and pharmacological activities of polysaccharides and their derivatives. Antioxidant activity is one of the properties of polysaccharides, particularly in terms of relief from skin aging. Polysaccharides such as agaric polysaccharides, *Lycium* polysaccharides, algal polysaccharides, lingzhi polysaccharides, and mushroom polysaccharides have been found to alleviate skin aging. Dietary polysaccharides have a positive effect on the improvement of aging skin. Mechanistically, oral polysaccharides enhance skin antioxidant enzyme activity, remove ROS, and reduce oxidative damage. They regulate the expression of Bcl–2, Bax, and Caspase–3 by activating Nrf2/ARE and other pathways, and inhibit apoptosis. Finally, polysaccharides inhibit collagen degradation by inhibiting the expression of enzymes such as MMP–1 and MMP–9, maintaining a stable collagen ratio, repairing skin structure, and maintaining skin moisture content [116,117,118,119,120,121].

### 4.4. Vitamins

Many vitamins have been tested for their antioxidant properties. They can reduce ROS in aging skin cells to low-activity molecules and reduce oxidative damage to key components of skin cells. Most research has focused on vitamins A, B (B3, B12), E, D, C, coenzyme Q10, and lipoic acid. Retinoids are the most common anti-aging drugs that have been used (such as retinoic acid prevents skin aging by regulating genes and MMPs) to treat and prevent photo-aging of the skin [97,98]. B vitamins have been shown to prevent skin aging mainly by preventing skin inflammation and pigmentation [99]. Vitamin C is a powerful antioxidant, its concentration in the skin is closely related to skin biological functions, and it is often used as a positive control for skin aging tests. It acts as an enzymatic factor and antioxidant to promote collagen synthesis and eliminate cellular ROS to relieve skin aging [100]. There seems to be a contradictory conclusion that vitamins D can alleviate skin photo-aging, because light not only synthesizes vitamins D but also causes skin aging. However, research shows that vitamin D can reduce DNA damage, inflammation, and photocarcinogenesis caused by ultraviolet rays, and thus, protect the skin [101]. A combination of vitamin E and C can help activate vitamins E, which protects skin against chemical stimuli and UV-induced irritation and damage by inhibiting lipid peroxidation in the skin [102,103]. Coenzyme Q10 is a vitamin-like substance widely present in meat foods, and its anti-aging function has been proven [122]. However, due to the instability, poor water solubility, and low utilization of vitamins during storage, vitamins are often used in combination with other antioxidant ingredients (such as collagen, astaxanthin, carotenoids, etc.) to enhance their anti-aging effects.

### 4.5. Fatty Acids

Lipids are an important part of the skin and are closely related to skin epidermal barrier function, membrane structure, internal environment balance, and damage repair. Skin aging is accompanied by a decrease in fat content, mainly due to a decrease in the ability of cells within the skin to synthesize and secrete fat [123]. Besides, the amount of dietary fat intake is closely related to the lipid composition of the body and skin tissues, and insufficient intake of essential fatty acids or abnormal fat metabolism leads to serious skin diseases [86,124]. Omega-3 and omega-6 polyunsaturated fatty acids play an important role as human skin barriers, and also have certain effects in the prevention and treatment of skin inflammation [104]. Oral olive oil can reduce skin aging induced by chronic psychological stress by acting on the NF-B NRF2 pathway in mice [125] (Table 2). Oral 7-MEGA^TM^ 500 (a product containing more than 50% palmitoleic acid containing fish oil, omega-7) has been shown to relieve UV-B and H_2_O_2_-induced skin oxidative stress, inflammation, aging, and promotes skin regeneration in mice [126,127]. The fatty acids extracted from *W. somnifera* seeds have good anti-inflammatory effects and exert an enhanced effect on psoriasis by reducing the release of pro-inflammatory factors (TNF-α and IL–6) [128]. The fermented fish oil protects skin aging by inhibiting PM_2.5_-induced ROS, MMP-s, and blocking the mitogen-activated protein kinase/activator protein 1 (MAPK/AP–1) pathway [129]. The effect of dietary and in vitro topical fatty acids on skin aging has been reviewed in detail by Wang and Wu [130].

### 4.6. Other Anti-Aging Nutrients

In addition to the aforementioned foodborne antioxidants that can be used as functional food ingredients to relieve skin aging, the combined use of different types of antioxidants has also been reported. Some studies have reported that dietary probiotics and their products can also alleviate skin aging. For example, probiotic fermentation can enhance the skin anti-photo-aging activity of *Agastache rugosa* leaves, and some probiotic extracts also have the potential to improve aging skin. [131,132]. There may be some foodborne antioxidants, which have not been mentioned in this article and need to be discussed subsequently.

**Table 2 nutrients-12-00870-t002:** Summary of the key in vivo studies investigating the potential effects of foodborne antioxidants in the skin aging.

References/Country/Study Type	Antioxidants/Source	Participants/Age	Induction Factors	Group/Dose/Time	Main Result	Main Conclusion
Effects of Oral Collagen Peptides on Skin Aging						
[133]/China/animal	High-, medium- and low-antioxidant peptides (HCP, MCP and LCP)/silver carp skin	KM mice/5 week (25 ± 2 g)	3UV-A + 1UV-B	Tg: HCP, MCP, LCP (0,200 mg/kg.bw.d)/0/1/2 weeks	1. ACPs significantly alleviated skin composition and antioxidant index abnormalities induced by UVs. 2. HCP has the best protection effect on skin photoaging, and the difference between MCP and LCP is not obvious.	ACPs have the potential to resist photoaging of the skin.
[134]/China/animal	Gelatin (SG)and gelatin hydrolysate (SGH)/ salmon skin	ICR male mice/ 20 to 22 g	UV-B	Tg: SG (100, 500 mg/kg.bw.d), SGH (100, 500 mg/kg.bw.d); Cg: Vc 100 mg/kg.bw.d/5 week	1. Antioxidant activity of SG and SGH is related to dose, molecular weight and amino acid composition. 2. SG and SGH alleviate oxidative damage by enhancing antioxidant enzyme activity and thymus index	SGH has the potential to be used as an antioxidant in health products and cosmetics.
[135]/Korea/animal	Collagen peptide (CP)/ tilapia scale	SKH–1 hairless mice/ 6 weeks old	UV-B	Tg:CP (0,500, 1000 mg/kg.bw.d) Cg: N–acetyl glucosamine (1000 mg/kg.bw.d)/ 9 weeks	1. Oral CP increased skin hydration, reduced wrinkle formation, changed the expression of HAS–1,–2, and maintained the stability of HA. 2. CP regulate the expression of skin moisturizing factor filagglutinin and total chain protein	CP can be used as a nutrient to relieve UV-B-induced skin wrinkles, dehydration and water loss.
[136]/Brazil/cell	Collagen Hydrolysate (CH)/cow	HDFs		Cg:CH (0.5, 1.0, 2.5 and 5.0 mg/mL)/48 h	1. CH regulates cell metabolism without cytotoxicity. 2. CH maintains intracellular protein stability by inhibiting the activity of MMP 1 and 2.	This CH has protective effects on skin cells and has the potential to become a food supplement.
[137]/Korea/clinical	Low-molecular-weight Collagen peptide (LMWCP)/ catfish’s skin	Women/40–60 years old	Age	Tg:LMWCP; 1000 mg/d. Cg: placebo; (0/6/12) weeks	Oral LMWCP protects photoaged skin by improving skin wrinkles, hydration and elasticity	LMWCP can be used as a functional food ingredient to relieve skin photoaging.
[138]/China/animal	Collagen hydrolysates (CHs)/Nile tilapia skin	ICRmice/38 ± 4 g, 9-month-old	Age	Tg:CHs (0%, 2.5%, 5%, 10%); Ng: weaned mice; Cg: (WC, 10% whey protein hydrolysates)/180 days	1. CHs significantly improves skin visual appearance, tissue structure and matrix homeostasis. 2. CHs alleviates oxidative stress by increasing skin antioxidant activity	CHs can be used as a functional nutritional food against skin aging, but its molecular mechanism is not clear.
[139]/China/animal	Elastin peptides (EH)/bovine arteries	Female mice/ (20 ± 2 g)	UV	Nc:vehicle-treated mice; Mg:vehicle-treated + UV. EH group:UV + EH (100 mg/kg.bw.d)/8 weeks	EH can significantly reduce UV-induced epidermal hyperplasia and fibroblast apoptosis, and increase skin hydroxyproline and water content	EH has the potential to prevent and regulate skin photoaging
[140]/China/animal	Collagen peptides (CPs)/silver carp skin	Mice/(8, 13-month-old (28 ± 2, 45 ± 5 g)	Age	Cg: young mice (normal saline); Tg: old mice (CPs: 400 mg/kg.bw.d); Mg: Old mice (normal saline)/2 months	1. CPs promotes skin collagen synthesis by regulating cytokines in skin and serum. 2. Intake of CPs inhibited platelet release.	CPs has the potential to be an anti-aging, anti- cancer and anti- cardiovascular health product
[141]/Canada/cell	Collagen peptides (CPs)/Chicken meat	HDFs cells/ human skin	DCF-DA	Tg: Two peptides, hydrolyzed by two enzymes (0, 2.5 mg/mL)/24 h	Two chicken collagen peptides have significant effects on inflammatory changes, oxidative stress, type I collagen synthesis, and cell proliferation in skin HDFs	CPs hydrolyzed by different enzymes have different protective and regulatory effects on skin fibroblasts
[142]/Canada/cell	Collagen peptides (CPs)/porcine/ bovine/tilapia/hen skin	HDFs/ human skin	UV-A	Tg: Four kinds of collagen peptides (0, 0.5, 1, 2, 4 mg/mL)/24 h	1. Bovine CH inhibits the MMP–1 production. 2. Tilapia CH promotes cell viability and type I collagen generation, while inhibiting ROS and MMP–3 generation. 3. Hen CH promotes collagen production and reduces ROS, MMP–1 and 9 generation and the expression of apoptotic genes.	Hen CH protects HDFs from UV-A-induced damage better than pigs, cattle and tilapia.
[143]/China/animal	High, low molecular weight collagen hydrolysates (HMCH/LMCH)/Silver Carp	Mice/5weeks (25 ± 2 g)	UV-A + UV-B	Tg1:UV + LMCH (HMCH)(50, 100, 200 mg/kg.bw.d)/6 weeks; Tg2:UV+LMCH (HMCH) (200 mg/kg.bw.d)/ 2 weeks	1. Both HMCH and LMCH increase skin components and antioxidant enzyme activity in skin and serum. 2. LMCH is more effective than HMCH. 3. Skin hydroxyproline, HA, and moisture content depend on peptide dose.	LMCH extracted from silver carp skin can be used as a dietary supplement to prevent skin aging.
[144]/Japan /clinical	High, low purity collagen hydrolysate (H-CP/L- CP)/fish gelatin	Female/(35–55 years old)	Age	H-CP group: 5 g/d; L-CP group: 5 g/d; Cp: placebo; 0/4/8 weeks.	H-CP is more significant than L-CP in improving facial skin moisture, elasticity, wrinkles, and roughness.	L-CP and H-CP are both effective dietary supplements to improve skin conditions.
[145] /Thailand /cell/animal	Collagen hydrolysate (HC)/Lates calcarifer skin	HDFs/human; Wistar rats (214 ± 26 g)		Mice Tg: (0,2000,5000 mg/kg.bw.d)/15d; Cell Tg: (50, 100, 150 and 200 µg/mL)/24 h.	1. Animal and cell experiments prove that HC is non-toxic. 2. HC can promote the growth of fibroblasts and the synthesis of cellular collagen, but not as effective as HC combined with VC.	Single HC or HC combined with VC can be used as nutritional health products for skin care.
[146]/China /cell	Gelatin hydrolysates (CGH)/Cod skin	HDF cells/ Mouse skin	UV-B	Tg: CGH (0, 0.001, 0.01, 0.1,1, 10) mg/mL /24 h.	1. CGH inhibits the expression of MMP–1 in fibroblasts induced by UV-B. 2. Purified MMP–1 inhibitory peptides have significant inhibitory effects on MMP–1, p-ER and p-p38.	CGH can be used as a functional supplement for skin care.
[147]/China /cell/animal	High, medium, low antioxidant peptide (HCP/ MCP/LCP)/Silver carp skin; Serum collagen peptides (SCP)/ rat serum	SD rat (8 week); ESF cells/skin	UV-A	Rats Tg (HCP, MCP and LCP)/(2.4 g/kg.bw.d)/2 h; Cell Tg: (SHCP, SMCP and SLCP)/(0, 50, 200 µM/mL)/24 h.	1. SCP is the active component of serum metabolites, which shows repair effect by removing ROS. 2. SCP promotes collagen synthesis and inhibits its degradation by activating TGF-β/Smad3 pathway and inhibiting the expression of AP–1 and MMP–1,3,SHCP is the best one.	CP promotes photoaging skin repair by activating the TGF- TGF/Smad pathway and inhibiting collagen reduction.
[148]/China /animal	Alcalase, Collagenase Collagen peptide (ACP/CCP)/ bovine bone	Mice/(8, 13-month-old (28 ± 2, 45 ± 5 g)	Age	Cg: young mice (normal saline); Tg: old mice/ACP (200, 400, and 800 mg/kg.bw.d), CCP (400 mg/kg.bw.d)/8 weeks	Oral CPs improve skin relaxation, increase collagen content and antioxidant enzyme activity, repair collagen fibers, and normalize the ratio of skin collagen. ACP is better than CCP.	CP can alleviate the chronological aging of the skin and has the potential to become an anti-aging functional food.
[149]/Korea /animal/cell	Collagen peptide NS (CPNS)/fish scale	HDF cells /human, Mice/8 weeks old (25–30 g)	UV-B	Cell Tg: CPNS (0, 50, 100, 250, 500 µg/mL)/24 h; Mice Tg: CPNS (300, 500 mg/kg.bw.d)/12weeks	1. CPNS treatment reduced the production of MMP–1 and increased the synthesis of type 1 procollagen in HFD cells. 2. Oral CPNS significantly reduced skin wrinkle formation, epidermal water loss, epidermal thickness, and increased hydration.	CPNS are a potential food supplement to prevent skin aging.
[150]/China /animal	Gelatin/Amur sturgeon swim bladder	Female SD rat/6 months old	Age	Cg (8% whey protein); Tg (8%, 4%, 2%)/12 months.	1. Oral administration of 3.85 g/kg.bw.d gelatin significantly improved skin histological structure and collagen ratio. 2. Skin antioxidant activity increased.	The gelatin improves the foundation for the development of anti-aging foods.
[110]/China /animal	Protein hydrolysate (WPH)/Walnut	SD rats/ 180–200 g	UV-A + UV-B	Cg:( distilled water); Tg: UV-R + WPH (0, 0.32, 0.98, 2.88 g/L)/18 weeks	1. WPH significantly enhances skin elasticity and promotes the biosynthesis of Col I, Hyp, and HA. 2. WPH inhibits MMP–1 activity and repairs skin damage. 3. WPH repair effect becomes dose dependent, high dose is best.	WPH has potential as a functional food ingredient against photoaging.
**Effects of Oral Polyphenol on Skin Aging**						
[151,152] /China/cell /animal	Polyphenol extract (HPE)/hawthorn	HDFs and HaCaT/human; mice/5–6 weeks old	UV-B	Cell Tg: HPE (0, 5, 10 µg/mL)/24 h; Mice Tg: HPE (0, 100, 300 mg/kg.bw.day)/12 weeks	1. HPE treatment can promote cell proliferation, increase intracellular collagen and reduce MMP–1 production. 2. Oral HPE reduces UV-B-induced skin damage by eliminating ROS, reducing DNA damage and inhibiting p53 expression.	HPE can be used as an anti-aging food or cosmetic ingredient.
[153]/Spain /clinical	Products rich in polyphenol (Nutroxsun^TM^)/rosemary and citrus	Adult female	UV-B + UV-A	Long-term: Nutroxsun^TM^ (250 mg/day)/ 2 weeks; Short-term: Nutroxsun^TM^ (100, 250 mg/day)/24, 48 h	1. Dietary Nutroxsun^TM^ reduces UV- induced skin changes, wrinkles and elasticity improvements. 2. The improvement effect between two doses is not obvious.	Long-term oral Nutroxsun^TM^ can be used as a nutritional supplement to improve skin conditions.
[154]/Korea /cell	Polyphenolic- rich extract (SSE and SSW)/*Spatholobus Suberectus* stem	HaCaT/Human skin	UV-B	Tg1: SSE (0, 3, 10, 30, 300 µg/mL); Tg2: SSW (0, 3, 10, 30, 300 µg/mL)/24 h	1. SSE and SSW inhibited ROS production and cell damage. 2. SSE repairs skin by upregulating the expression of enzymes and proteins in cells, blocking UV-B-induced MAPKs phosphorylation and its downstream transcription factor.	SSE can be used as a natural biomaterial to inhibit UV-B-induced photoaging.
[155]/China /animal	Rambutan peel phenolics (RPP)/*Nephelium lappaceum*; Leu-Ser-Gly-Tyr-Gly-Pro (LSGYGP)/synthetic	Male BALB/c nude mice/20–22 g	UV-B	Single group: RPP (100 mg/kg.bw. d), SGYGP (100 mg/kg.bw.d); Composite group: (50 RPP+ 50 LSGYGP) mg/kg.bw.d, (100 RPP + 100 LSGYGP)mg/kg.bw.d/10 weeks	1. RPP and LSGYGP improve skin biochemical indicators, tissue structure and collagen levels. 2. RPP enhances the regulation of oxidative stress and inflammatory factor levels. 3. LSGYGP significantly affects skin collagen and HA content.	Oral RPP and LSGYGP can alleviate UV-B- induced skin aging.
[156]/Korea /animal	Polyphenols/Flavonoid hesperidin exerts	Male hairless mice/6- week-old	UV-B	Cg: water; Tg: UV-B + hesperidin (0, 100 mg/kg.bw.d)/12 weeks	1. Oral hesperidin inhibited UV-B-induced skin thickening and wrinkle formation. 2. Hesperidin inhibited UV-B-induced expression of MMP–9 and cytokines, and protected collagen fiber loss.	Oral hesperidin regulates MMP–9 expression by inhibiting MAPK- dependent signaling pathways to relieve skin photo-aging.
[157]/Korea /cell	Polyphenols/3,5,6,7,8,3,4-heptam-ethoxy flavone (HMF)/C. unshiu peels	HDFn cells/human dermal	UV-B	HMF (0, 50, 100, 200 µg/mL)/24 h	1. HMF protects UV-induced HDFn cell damage by inhibiting MMP–1 expression through phosphorylated MAPK signals. 2. HMF regulates the expression of Smad3 and Smad7 proteins in a dose-dependent manner.	HMF has the potential to be an anti-aging cosmetic or food supplement.
[158]/Korea /cell	Polyphenols/ Tectorigenin/ Belamcanda chinensis L	HaCaT cells/human	UV-B	Tg: Tectorigenin (0, 0.1, 1,10 µM); Cg: VC (200 µM)/24 h	1. Tectorigenin lowers ROS levels by increasing intracellular antioxidant enzymes. 2. Tectorigenin reduces mmp–1 and inhibits collagen degradation. 3. Tectorigenin inhibits apoptosis by regulating the levels of caspase–3 and bcl–2 related proteins.	Tectorigenin alleviates skin damage by inhibiting UV-B-induced cellular oxidation, apoptosis and collagen degradation.
**Effects of Oral Polysaccharides on Skin Aging**						
[116]/China /animal	Polysaccharides (TP)/T. fuciformis	SD rats/6~7 weeks old (180–220 g)	UV-A + UV-B	Cg: no irradiation; Tg group: UV + TP (0, 100, 200, 300 mg/kg.bw.d)/12 weeks	Oral TP can alleviate UV-induced skin structural changes, repair collagen damage, maintain the I/III collagen ratio and enhance skin antioxidant enzyme activity.	TP has the potential to become a skin-protective functional food additive.
[117]/Korea /cell	Polysaccharide (HFPS)/Hizikia fusiforme	HDF cells	UV-B	Cg: no irradiation; Tp: UV + HFPS (0, 25, 50, 100 µg/mL)/24 h	1. HFPS significantly reduces cell ROS and increases the pure activity rate. 2.HFPS inhibits UV-induced skin damage by regulating NF-κB, ap–1 and MAPKs signaling pathways.	HFPS has a strong anti-ultraviolet effect and is a potential pharmaceutical, food, and cosmetic ingredient.
[118]/China /cell	Polysaccharide (LBP)/Lycium barbarum	HaCaT cells	UV-B	Tg1: LBP (0, 50, 100, 300, 600, 1500, 3000 µg/mL) 24 h; Tp2: UV-B + LBP (0, 300 µg/mL)/24 h	LBP mainly eliminates ROS and reduces DNA damage. In part, the Nrf2/ARE pathway is activated to inhibit the p38 MAP pathway, thereby inhibiting the activation of caspase–3 and the expression of mmp–9 to protect the aging cells.	LBP may be used as a protective agent or food additive against skin oxidative damage.
[119]/China /cell	Polysaccharide (GL-PS)/Ganoderma lucidum	Fibroblast/men foreskin	UV-B	Tg: UV-B + GL-PS (0, 10, 20, 40 µg/mL) 24 h; Tg: no UV-B and GL-PS/24 h	After GL-PS treatment, cell activity increased, senescent cells decreased, CICP protein expression increased, MMP–1 protein expression decreased, and cell ROS level decreased.	GL-PS protects UV-B- induced cell photoaging by eliminating intracellular ROS, which will provide strategies for subsequent studies.
[120]/China /animal	Polysaccharide (SFP)/Sargassum fusiforme	Female KM mice/7 weeks old (20–25 g)	UV-B	Cg: UV-B + sodium hyaluronate (400 mg/kg. bw/d); SFP Tg: UV-B + SFP (0, 200, 400, 600 mg/kg.bw/d)/9 weeks	1. SFP regulates mouse chest, spleen index and skin water content. 2. SFP increases skin antioxidant enzyme activity, reduces ROS, and reduces oxidative damage.3.SFP inhibits MMP–1 and 9 levels in the skin.	SFP can be a potential functional food additive for skin protection.
[121]/China /cell/animal	Purified, crude polysaccharide (TLH–3,TLH)/ Tricholoma lobayense	HELF cells/human; Mice/8 weeks (23 ± 2 g)	t-BHP/D-galactose	Cell Tg: TLH–3 (0, 50, 100, 200, 400 µg/mL), Pc: Vc (50 ug/mL)/24 h; Mice Tg: TLH–3 and TLH (200 mg/kg. bw/d),Pc: Vc (100 mg/kg. bw/d)/5 weeks	1. TLH–3 relieves cell senescence by regulating the expression of bcl–2, bax, caspase–3 proteins, inhibiting senescence-related enzyme levels. 2. TLH–3 reduced skin pathological lesions by reducing IL–6, LPF, AGEs, and enhanced MAO activity.	TLH–3 is an active polysaccharide that protects cells and mice from oxidative stress aging.
**Effects of Oral Vitamins on Skin Aging**						
[159]/China /cell	Vitamin Coenzyme Q_10_ (CoQ_10_)	ESF and HaCaT cells/ Human	UV-A, UV-B	Cg: ESF, HaCat (CoQ_10_ (0, 0.5, 1, 2 µM))/24 h; Tg: ESF, HaCat (UV-A or UV-B + CoQ_10_ (0, 1, 5, 10 µM))/24 h	1. CoQ_10_ treatment promoted ESF cell proliferation, type IV collagen and elastin gene expression. 2. CoQ_10_ treatment inhibited UV-induced IL–1a production in HaCaT cells.	CoQ_10_ has anti-aging effect on chronological aging and photo-aging and can be used in food and cosmetics.
[122]/Japan /clinical	VC, VE, and Astaxanthin (AX)	Female/(mean age 37.26 years)	-	Tg1:AX (6 mg) + VC (1000 mg) + VE (10 mg)/d; Tg2:VC (1000 mg) + VE (10 mg)/d/20 weeks	Tg 1 significantly improved skin moisture content, skin elasticity and wrinkles; Tg 2 did not improve the skin significantly.	Oral formulations containing astaxanthin and vitamin C and E have skin-improving effects.
[160]/Iran /animal	Silymarin, Vitamin C	Balb/C mice/6 weeks old (30 ± 2 g)	UV-B	Cg: Silymarin (100),VC (40 mg/kg.bw/d)/; Tg: UV-B + Silymarin (0, 100 mg/kg.bw/d), UV-B + VC: (0, 40 mg/kg.bw/d)/4 weeks.	Oral VC enhances skin antioxidant enzyme activity, reduces skin wrinkle formation and thickness increase in mice induced by UV-B.	Salicylic acid and vitamin C can be used as food or cosmetic ingredients to resist skin photo-aging.
[161]/Korea /cell	Niacinamide (NIA)	HaCaT/ human	PM_2.5_	Cg: NIA (0, 12.5, 25, 50, 100, or 200 µM); Tg: NIA (0, 12.5, 25, 50, 100, or 200 µM) + PM_2.5_ (50 µM)/24 h	NIA treatment can inhibit the oxidation of lipid, protein, DNA and other molecules induced by PM _2.5_, as well as inhibit apoptosis and ROS production.	NIA protects cells from PM _2.5_-induced oxidative stress and cell damage.
**Effects of Oral Fatty Acids on Skin Aging**						
[125]/Brazil /animal	0live oil	Swiss mice/8–12 weeks age	Rotational stress	Stress group: stress + olive (1.5 g/kg.bw. d), Cg: olive (1.5 g/kg. bw/d)/29 d	Olive inhibited skin ROS, lipid peroxidation, protein carbonylation, phenolamine synthesis, MMP–8 expression and promotes collagen deposition in mice through NF-κB and NRF2 pathways.	Oral administration of olive oil can reduce mice skin aging induced by stress.
[126]/Korea /animal	7-MEGA^TM^500/> 50% of palmitoleic acid containing fish oil, omega–7	H–1 mice/5-week old (18–20 g)	UV-B	Cg: 30% EtOH; Tp: 7-MEGA^TM^500 (50, 100, 200 mg/kg.bw/d)/4 weeks	1.7-MEGA^TM^ 500 improves skin histological indicators and significantly down-regulated the expression levels of MMP–3 and c-jun genes and proteins in the skin.	7-MEGA^TM^500 can alleviate UV-B induced skin photoaging in mice
[129]/Korea /cell	Fermented Fish Oil (FFO)	HaCaT/ human	PM_2.5_	Cg: PM_2.5_; Tp:PM_2.5_ + FFO (0, 20 µg/mL)/24 h	FFO can inhibit PM _2.5_-induced intracellular ROS, Ca ^2+^ levels and MMPs–1,2,9 production, and block the MAPK/AP–1 pathway.	FFO can alleviate PM _2.5_ induced skin aging.
[162]/Japan /animal	Coconut oil	Female mice/(6 weeks old)	DNFB	Cg: Coconut or soybean oil (4%)/2 months; Tg: Coconut or soybean oil (4%)/after 2 months + DNFB	Oral coconut oil improves BDFB-induced skin inflammation in mice. Mechanistically related to elevated mead acid in serum inhibiting directional migration of neutrophils.	Dietary coconut oil improved skin contact allergies in mice by producing midic acid.

Cg = control group; Tg = test group; Ng = normal group; Mg = model group; mg/kg.bw.d = mg/kg. body weight/day; DNFB = 1-fluoro-2,4-dinitro-benzene; HaCaT = Human skin epidermal keratinocytes; t-BHP = tert-butyl hydroperoxide; DCF-DA = dichlorofluorescein diacetate; HAS-1,-2 = hyaluronic acid synthases1and 2; MAO = monoamine oxidase; Co Q10 = Coenzyme Q10; LPF = lipofuscin pigment; NF-κB = nuclear factor kappa B; IL-6 = Interleukin-6; IL-1 = Interleukin-1; NRF2 = nuclear factor erythroid-2p45-related factor2; ARE = antioxidant response element; p38 MAP = p38 mitogen-activated protein; MAPKs = Mitogen-activated protein kinases; TGF-β/Smad3 = Transforming growth factor-β/Recombinant Human Mothers Against Decapentaplegic Homolog 3; p-ERK = phosphorylated extracellular regulated kinase; p-p38 = phospho-p38.

## 5. Conclusions and Prospects

Skin aging is a complex and long biological process, which is affected by genetic and environmental factors. Although stem cell transplantation, injection of hyaluronic acid, and retinoic acid have certain therapeutic effects, each method has corresponding disadvantages. With the improvement of people’s requirements for the effectiveness, safety, and durability of treatment methods, prevention, and relief from skin aging through dietary management have become an inevitable trend. Therefore, after analyzing and summarizing relevant literature, we draw the following key conclusions:

People’s current understanding of diet to improve skin aging is still insufficient. While it is difficult for us to accurately define what is a healthy diet, and to quantify the relationship between diet and skin aging that convince the public, it is difficult for them to change their original lifestyle and diet, even if people have such knowledge.The issues of accurately quantifying the skin improvement effect of each nutrient intake, and the negative effects of smoking, drinking, grilling, etc., on skin aging still need to be addressed.The functional anti-aging ingredients in food mainly relieve skin aging in three ways. First, anti-aging ingredients (such as protein peptides and essential fatty acids) enter the skin as a precursor after digestion and absorption and participate in the synthesis and metabolism of skin components. Second, anti-aging ingredients relieve skin oxidative damage by removing cellular ROS and enhancing antioxidant enzyme activity. Third, the anti-aging component acts as an enzymatic factor, and regulates the expression of enzymes such as MMPs and AP–1, inhibiting the degradation of skin components and maintaining the integrity of the skin structure.The limitations of foodborne antioxidants such as unstable storage, low skin bioavailability, and poor solubility, can be improved by chemical modification, collagen drug delivery, and a combination of supplements.Only oral supplementation is not enough to improve the skin. The combination of oral and external skin penetration should be the safest and the most effective way to improve skin aging.Diet causes skin aging or improves skin aging and is difficult to simply apply to clinical research. While on the one hand, there is an ethical controversy, on the other hand, the experimental period is too long to control the diet of volunteers for a single, long duration, and the uniformity of clinical experimental conditions is not guaranteed, resulting in vague experimental results and insufficient credibility.Improvement in skin aging through diet should not be rushed, because skin aging caused by diet and improvement of aging performance by diet are long-term processes. There is also the problem of metabolic processing of food and nutrients until they reach the skin. They have to travel a long way, and there is still a lot to study in this process.

## Figures and Tables

**Figure 1 nutrients-12-00870-f001:**
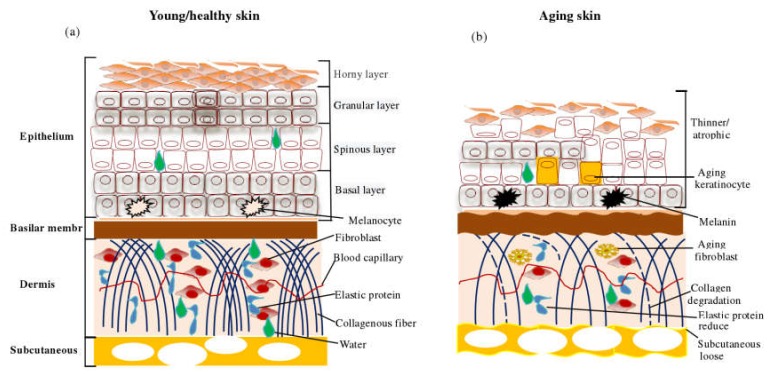
(**a**) Schematic diagram of skin structure, (**b**) Schematic diagram of skin structure after aging. This picture is a comparison of the changes between young skin and aging skin.

**Figure 2 nutrients-12-00870-f002:**
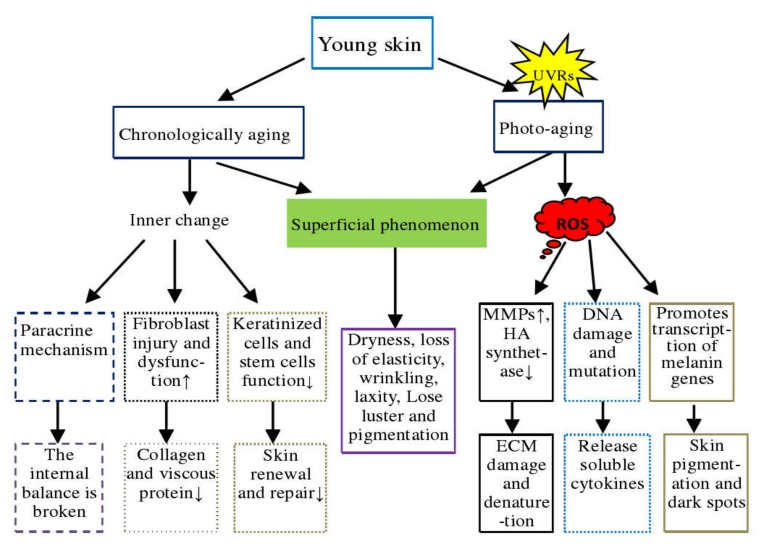
Comparison of chronological aging and photo-aging of the skin. In the figure, "↑" means "rise or increase"; "↓" means "fall or decrease"; DNA= deoxyribonucleic acid; ECM= extracellular matrix; UVRs = ultraviolet radiations; ROS= reactive oxygen species; MMPs = Matrix metalloproteinases; HA= Hyaluronic acid.

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
