# Peer review of "Diet and Skin Aging—From the Perspective of Food Nutrition"

_nutrients, 2020, doi:10.3390/nu12030870_

Round 1

Reviewer 1 Report

In the review “Diet and skin aging – From the perspective of food nutrition” the author reviewed the state of art on the effect of diet on the evolution of skin aging processes. During the review, Several important aspects related to skin (structure, aging process mechanisms, the effect of diet on skin aging) were addressed along with the manuscript. The review contributes to the field of skin nutraceuticals by gathering valuable information for future research. The paper is well written; however, some minor English typos are found.

Authors should address some minor remarks:

  • Figure 1 – please increase the overall quality of figure 1
  • Figure 2 – please increase the overall quality of figure 1

Author Response

Thanks for the expert's advice. I have tried my best to make quality modifications to the pictures in the article. I hope it can be approved. I have looked over the words in the article.

Reviewer 2 Report

This manuscript, entitled “Diet and skin aging – From the perspective of food nutrition” written by Cao, reviews comprehensively relationship diet and skin aging. This paper is well-written and will be able to provide valuable information for researchers. My evaluation is that this paper is publishable with minor revisions about reference styles.

Author Response

Thanks for the expert's advice. I haven't made any major revisions to the references in this paper, and I will revise them according to the comments of the proofreader in the future.

Reviewer 3 Report

Interesting and comprehensive paper. Some minor comments:

line 53: maybe mention skin aging exposome, Krutmann et al. 2017 J Dermatol Sci

Figure 1: remove the hair in both panels. It does not contribute information and is only confusing.

line 74: instead of 'race', rather use skin type or ethnicity

line 74: 'skin parts': do you mean skin site?

line 77: 'even no tissue': can you give a reference for this?

line 160: instead of 'race', rather use ethnicity or ethnic group

line 232: 'skin cuticle' is not really used. do you mean stratum corneum?

Section 4.4.: at the end you mention 'astaxanthin': maybe also mention carotenoids in general. They are somewhat missing in the paper.

line 446-450: maybe add that there is also the problem of metabolic processing of food and nutrients until they reach the skin. They have to travel a long way.

Author Response

Thanks to the expert for reviewing the manuscript.All changes are marked in red in the article.

line 53: I have paid attention to the articles listed by reviewers in this paper, thank you very much.

Figure 1: "Hair" has been deleted from the picture1.

line 74: Having replaced the” race” with the “ethnicity”

line 74: Yes, the article has changed to "skin site"

line 77: Thanks to the expert opinion, it is my language writing problem, which has been changed to "disorganized".

line 160:Ok, it has been corrected in the article

line 232: Yes, it has been revised in the article. Thanks for the expert's opinion.

Section 4.4 simply refers to 'astaxanthin' and so on.This part to join the 'carotenoids.However, due to the structure of the article, it cannot be described in detail.

line 446-450: Thanks to the expert's comments, which have been added at the end.